# Axillary Lymphadenopathy on Ultrasound after COVID-19 Vaccination and Its Influencing Factors: A Single-Center Study

**DOI:** 10.3390/jcm11010238

**Published:** 2022-01-03

**Authors:** Ji Yeon Park, Ji Young Lee, Seong Yoon Yi

**Affiliations:** 1Department of Radiology, Inje University Ilsan Paik Hospital, Goyang 10380, Korea; zzzz3@hanmail.net; 2Division of Hematology-Oncology, Department of Internal Medicine, Inje University Ilsan Paik Hospital, Goyang 10380, Korea

**Keywords:** COVID-19, vaccine, axilla, lymphadenopathy, ultrasound

## Abstract

Purpose: This study aimed to assess the incidence of axillary lymphadenopathy on ultrasound after COVID-19 vaccination and to investigate the factors affecting lymphadenopathy. Methods: We evaluated patients who had received a COVID-19 vaccination within 12 weeks before an ultrasound examination between August and October 2021. The incidence of vaccine-related ipsilateral axillary lymphadenopathy was evaluated using ultrasound. Age, sex, presence of axillary symptoms, injection site, vaccine type, interval from vaccination, and dose were compared between the groups with and without axillary lymphadenopathy. Results: We included 413 patients, 202 (49%) of whom showed axillary lymphadenopathy on ultrasound after COVID-19 vaccination. Age, interval from vaccine, vaccine brand, vaccine type, dose, and symptom were significantly different between the lymphadenopathy and non-lymphadenopathy groups (*p* < 0.001), while the injection site and sex were not. Receiving an mRNA vaccine was the most important factor for axillary lymphadenopathy (*p* < 0.001), followed by intervals of 1–14 (*p* < 0.001) and 15–28 days (*p* < 0.001), younger age (*p* = 0.006), and first dose (*p* = 0.045). Conclusion: COVID-19 vaccine-related axillary lymphadenopathy on ultrasound is common. mRNA type, an interval of 4 weeks, younger age, and first dose were the important factors. Breast clinicians should be well aware of these side effects when performing imaging examinations and provide accurate information to patients.

## 1. Introduction

In conjunction with the massive administration of coronavirus disease 2019 (COVID-19) vaccines, we observed an unusual increase in the number of imaging-detected unilateral axillary lymphadenopathy cases at our breast imaging clinic. Vaccine-associated lymphadenopathy has also been recently reported as a frequent imaging finding after COVID-19 vaccination [1,2,3]. In clinical trials of the Moderna vaccine, axillary swelling or tenderness was reported in 11.6% and 16.0% of patients following the first and the second dose, retrospectively. Further, clinically detected lymphadenopathy was reported in 1.1% of participants within 2–4 days after vaccination [4]. For the Pfizer-BioNTech vaccine, the rate of lymphadenopathy was 0.3% [5,6]. For the AstraZeneca vaccine, lymphadenopathy was known to be a rare event that may affect up to one in 100 people [7]. These reported rates were based on clinical assessments, such as physical examination.

The incidence of axillary lymphadenopathy after COVID-19 vaccination on mammography is reportedly 3%, and the presence of symptoms and a shorter time after vaccination were associated with lymphadenopathy on mammography [8]. However, the incidence of axillary lymphadenopathy on ultrasound following COVID-19 vaccination has not yet been reported in a large number of cases, and only case series have been reported [9,10]. Further, although studies have been conducted, these only focused on sonographic findings of axillary lymphadenopathy, and not on the incidence or factors associated with vaccine-related lymphadenopathy [11,12].

To our knowledge, no study has investigated the incidence and associated factors of COVID-19 vaccine-related axillary lymphadenopathy on ultrasound among patients who received the Pfizer-BioNTech, Moderna, or AstraZeneca vaccine.

As COVID-19 vaccination is proceeding rapidly, radiologists are expected to increasingly encounter lymphadenopathy detected by breast imaging [9,13]. In the COVID-19 pandemic era, necessary imaging at breast clinics should no longer be postponed due to prior vaccination because the number of advanced-stage breast cancer patients are projected to increase after the COVID-19 pandemic [14]. Therefore, knowledge on the incidence of lymphadenopathy associated with the vaccine type, interval, and dose is important to manage patients visiting breast clinics.

Therefore, this study aimed to assess the incidence of axillary lymphadenopathy on ultrasound after COVID-19 vaccination and to investigate the factors associated with lymphadenopathy.

## 2. Materials and Methods

### 2.1. Study Design and Patients

The institutional review board approved this retrospective study and waived the needs for written informed consent. 

A total of 1299 patients underwent breast–axilla ultrasonography at our institution between August and October 2021. Among them, we selected 574 patients who received COVID-19 vaccination within 12 weeks before ultrasound examination. We excluded patients with uncertain vaccine type or date of vaccination (*n* = 40), any malignancy (*n* = 115), and ipsilateral mastitis (*n* = 4). Further, one patient who received the Janssen vaccine and one patient who received three doses of the Pfizer vaccine were excluded because of their small number, which might have been difficult to analyze statistically (Figure 1). 

### 2.2. Ultrasound Technique

One of two breast radiologists with 10–15 years of experience in breast and axillary imaging performed ultrasound examination using an Aixplorer (SuperSonic Imagine, Aix-en-Provence, France) or Aplio I800 (Canon Medical Systems Corporation, Tokyo, Japan) system with a 14–18 MHz linear transducer. The presence of abnormal axillar lymphadenopathy was retrospectively reviewed by two radiologists who were blinded to the patients’ clinical information. Abnormal lymphadenopathy was defined as enlarged lymph node with cortical thickening >3 mm with or without a preserved fatty hilum on ultrasound. Any discrepancies were resolved by consensus.

### 2.3. Data Collection

Data on the vaccine brand (Oxford-AstraZeneca, Pfizer-BioNTech, or Moderna), vaccine type (messenger RNA (mRNA) or vector), interval from vaccine (interval between ultrasound and vaccine), dose (first or second), injection site (left or right arm), and symptoms including axillary pain or swelling were collected from the medical records.

### 2.4. Statistical Analysis

Chi-square and Fisher’s exact tests were used to compare categorical variables, while Student’s t-test was used for continuous variables. Multiple linear regression analysis was used to evaluate the important factor associated with the lymphadenopathy. Statistical analyses were performed using SPSS 25.0 (IBM, Armonk, NY, USA). A value of *p* < 0.05 was considered significant. 

## 3. Results

A total of 413 patients (mean age ±standard deviation: 48 ± 12 years, range: 17–80 years, sex: 403 women and 10 men) were included in this study. In total, 389 (94%) and 24 (6%) patients were injected COVID-19 vaccination to the left and to the right arm, respectively. The mean interval from vaccination was 26 ± 18 (range: 1–82) days. A total of 121 (29%) patients received vaccination within 2 weeks before the ultrasound and 252 (61%) within 4 weeks. There were 330 (80%), 64 (15%), and 19 (5%) patients who received the Pfizer, AstraZeneca, and Moderna vaccines, respectively. A total of 156 (38%) and 257 (62%) patients had received the first and second dose, respectively. In total, 202 (49%) patients had axillary lymphadenopathy, while eight (4%) had symptoms, such as axillary pain or swelling (Table 1).

Table 2 and Figure 2 show the comparison of characteristics between the lymphadenopathy and non-lymphadenopathy groups. 

The lymphadenopathy group was significantly younger than non-lymphadenopathy (44 ± 11 vs. 52 ± 10 years, *p* < 0.001). At a cutoff age of 40 years, younger patients (i.e., those aged ≤40 years) showed a significantly higher rate of lymphadenopathy than the older patients (*p* < 0.001). The interval from vaccination was significantly shorter in the lymphadenopathy group (20 ± 14 vs. 32 ± 19 days, *p* < 0.001). In the lymphadenopathy group, the mean interval from vaccine was 19, 23, and 42 days for Pfizer, Moderna, and AstraZeneca, respectively. In the lymphadenopathy group, the rate of interval group was 38%, 41%, and 21% for interval from vaccination of 1–14, 15–28, and ≥29 days, respectively, and significantly different from those in the non-lymphadenopathy group (21%, 23%, and 56%, *p* < 0.001). Overall, 55%, 8%, and 79% of patients who received the Pfizer, AstraZeneca, and Moderna vaccines, respectively, developed lymphadenopathy (*p* < 0.001) (Figure 3, Figure 4 and Figure 5).

For vaccine type, 56% and 7.8% of patients who received mRNA and vector vaccines, respectively, developed lymphadenopathy (*p* < 0.001). Concerning the number of vaccines, 63% and 41% of those who had received their first and second doses showed lymphadenopathy (*p* < 0.001). However, sex and injection site were not significantly different between the two groups (*p* = 0.106 and 0.341). 

Multiple linear regression analysis showed that mRNA vaccine (odds ratio (OR) = 6.7, *p* < 0.001), interval of 1–14 days (OR = 4.3, *p* < 0.001), interval of 15–28 days (OR = 4.3, *p* < 0.001), younger age (≤ 40 years, OR = 2.2, *p* = 0.006), and first vaccination (OR = 1.6, *p* = 0.045) were independent factors associated with lymphadenopathy (Table 3).

## 4. Discussion

The current study found a high incidence of COVID-19 vaccine-induced axillary lymphadenopathy on ultrasound (49%), which was higher than the self-reported axillary swelling in previous COVID-19 vaccine trials (16%) [4]. We also showed that mRNA vaccine, interval of 1–28 days, younger age (≤40 years), and first vaccination independently influenced the occurrence of axillary lymphadenopathy on ultrasound.

Lymph node enlargement following vaccination is related to the accumulation of locally activated antigens at the injection site and later migration to draining nodes [15]. Previous literature shows that the conventional vaccines such as H1N1 influenza, smallpox, measles, Bacille Calmette–Guerin, and human papillomavirus vaccines can cause infrequent axillary lymphadenopathy [16,17,18,19,20].

A study comparing the immunogenicity between mRNA and vector vaccines against SARS-CoV-2 revealed spike-binding antibody and neutralizing antibody levels were higher in mRNA-vaccinated subjects. Meanwhile, there were no significant differences in antigen-specific B and T cell responses [21]. These results may cause difference in the incidence of ultrasound-detected axillary lymphadenopathy between mRNA and vector vaccines. In this study, the incidence of axillary lymphadenopathy on ultrasound was higher in individuals receiving mRNA than in those receiving vector vaccines.

The Society of Breast Imaging recommends a short-term follow-up exam in 4–12 weeks following the second vaccine dose for appropriate diagnostic work up for unilateral axillary lymphadenopathy and scheduling screening exams prior to the first dose of a COVID-19 vaccination or 4–6 weeks following the second dose [22]. The European Society of Breast Imaging also recommends that ultrasonography should be performed in cases of axillary lymphadenopathy > 12 weeks after vaccination in patients without a history of breast cancer [23]. According to these recommendations, we included only patients who underwent ultrasound within 12 weeks after vaccination. The mean interval from vaccination in the lymphadenopathy group was 20 days and was significantly shorter than that in the non-lymphadenopathy group (32 days). Further, 93% of ultrasound-detected lymphadenopathy was observed within 6 weeks after vaccination. A literature review of 68 cases found that 97% of imaging detected lymphadenopathy after a COVID-19 vaccination occurred from the first day to 4 weeks after vaccination, although lymphadenopathy remained after 6 weeks of vaccination in two cases [2]. Granata et al. evaluated 18 patients and found that axillary lymphadenopathy persisted in one patient at 73 days after Pfizer vaccination [12]. Eshet et al. reported that fluorodeoxyglucose uptake in lymphadenopathy persists between 7 and 10 weeks after the second dose of Pfizer vaccine in 29% of patients [24]. In our study, 7% of ultrasound-detected lymphadenopathy occurred ≥6 weeks after vaccination, and one patient showed lymphadenopathy 80 days after Pfizer vaccination. This patient underwent a core needle biopsy of the axillary lymph node, and the pathologic result was reactive hyperplasia. This suggests that vaccine-related lymphadenopathy may persist for ≥6 weeks after vaccination. 

The current study also found that age is a significant influencing factor of axillary lymphadenopathy after COVID-19 vaccination. Particularly, younger patients (≤40 years) showed a higher incidence of lymphadenopathy on ultrasound. This may be supported by findings of lower serum neutralization and antibody levels in older adults receiving the Pfizer-BioNTech vaccine than in younger adults [25,26]. The incidence of lymphadenopathy was also significantly higher after the first than after the second dose. This is supported by a previously published study reporting a significantly higher axillary lymph node response to vaccination in patients who were not previously infected by SARS-CoV-2 [27]. It seems logical that protein presentation caused by antigen-presenting cells may be less involved in the second clonal expansion at the local site because it has already occurred in other local areas of patients previously exposed during the infectious process [27]. The majority of axillary lymphadenopathy (96%) in the current study was subclinical and detected by ultrasound. This was consistent with the results of another study wherein 97.5% of lymphadenopathy in 163 cases in a breast imaging clinic were only detected by imaging such as ultrasound, magnetic resonance imaging, or mammography [1].

Our study had some limitations. First, this was a retrospective study performed in a single center, thus limiting the generalizability of the results. Second, the number of male patients was very small; therefore, the incidence of lymphadenopathy in men may not have been accurately evaluated. Third, the number of patients who received AstraZeneca and Moderna vaccines was also small. In our country, AstraZeneca vaccination started earlier than with Pfizer, and Moderna even later. Therefore, most of the patients included in this study had received the Pfizer vaccine. Fourth, we did not have information on follow-up ultrasounds. Further studies are needed to evaluate the improvement in vaccine-related lymphadenopathy on follow-up ultrasounds. Fifth, we did not compare lymphadenopathy between patients with and without a vaccination.

## 5. Conclusions

COVID-19 vaccine-induced axillary lymphadenopathy detected by ultrasound is common. Especially, mRNA vaccine type, interval of 4 weeks from vaccination, younger age, and first dose are independent factors that influence the occurrence of lymphadenopathy. Physicians and radiologists should be familiar with accurate information concerning COVID-19 vaccine-related lymphadenopathy to manage patients in breast clinics.

## Figures and Tables

**Figure 1 jcm-11-00238-f001:**
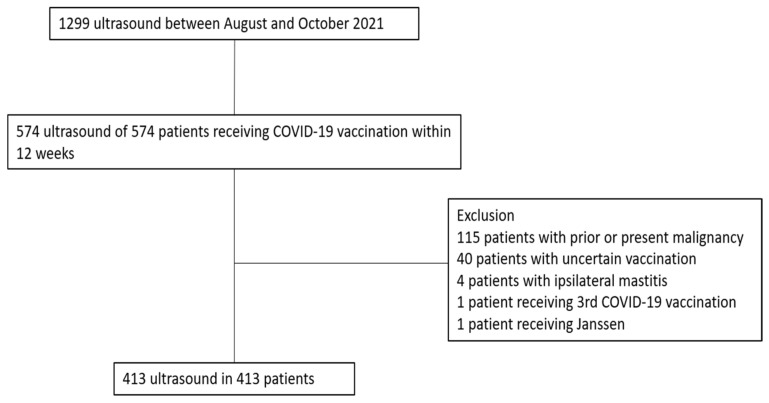
Flow chart of the study population.

**Figure 2 jcm-11-00238-f002:**
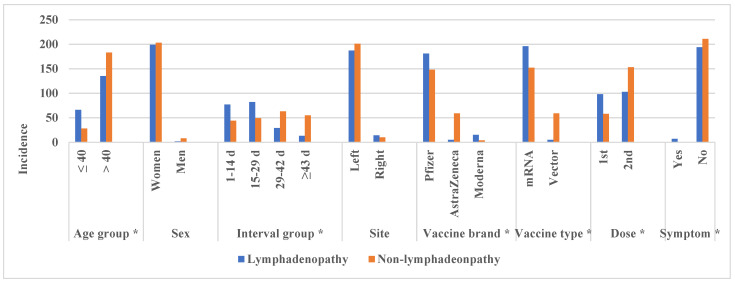
Histogram comparing the characteristics between the lymphadenopathy and non-lymphadenopathy groups. *: statistically significant (*p* < 0.05).

**Figure 3 jcm-11-00238-f003:**
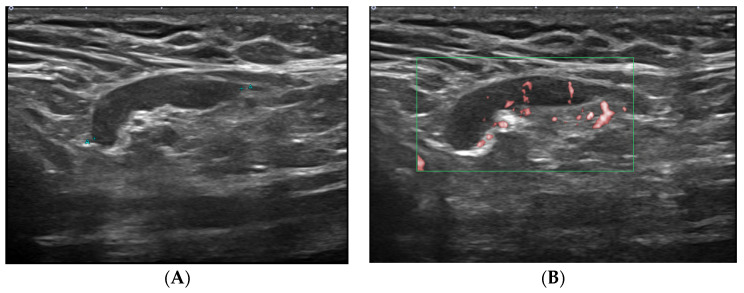
A 26-year-old woman with axillary lymphadenopathy on ultrasound 15 days after the first Pfizer vaccination in the left arm. (**A**) Gray scale ultrasound shows a 2.1 cm enlarged lymph node with 0.5 cm cortical thickening and preserved fatty hilum in the left axilla. (**B**) Doppler study shows hilar vascularity of lymph node in the left axilla.

**Figure 4 jcm-11-00238-f004:**
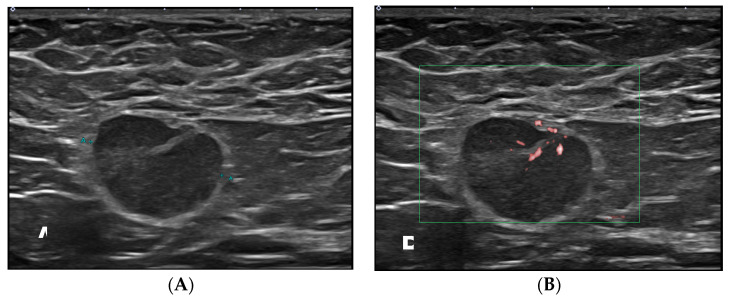
A 37-year-old woman with axillary lymphadenopathy on ultrasound 15 days after the first Moderna vaccination in the left arm. (**A**) Gray scale ultrasound shows a 1.8 cm round enlarged lymph node with an obliterated fatty hilum in the left axilla. (**B**) Doppler study shows hilar vascularity of lymph node in the left axilla.

**Figure 5 jcm-11-00238-f005:**
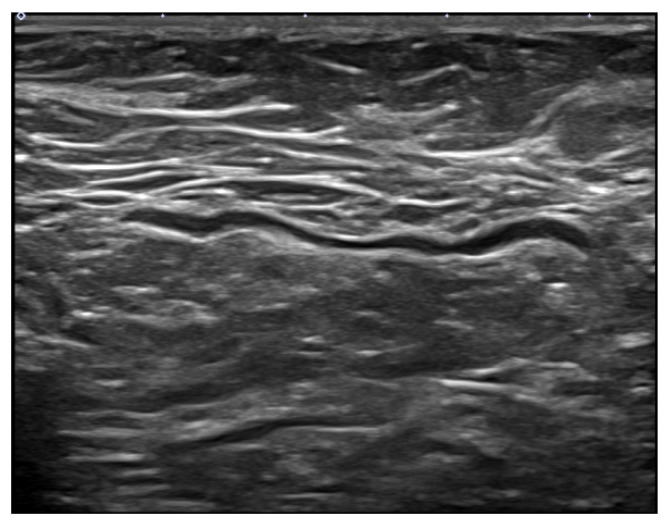
A 61-year-old woman without axillary lymphadenopathy on ultrasound 55 days after the second AstraZeneca vaccination in the left arm. Gray scale ultrasound shows a normal lymph node with 0.1 cm even cortex in the left axilla.

**Table 1 jcm-11-00238-t001:** Demographics and clinical characteristics of 413 patients with COVID-19 vaccination.

Characteristics	No. of Patients (*n* = 413)
Mean age (years) *	48 ± 12 (17–80)
Sex (Women/Men)	403:10 (98:2)
Mean interval from vaccination (days) *	26 ± 18 (1–82)
Site	
Left arm	389 (94)
Right arm	24 (6)
Vaccine brand	
Pfizer	330 (80)
AstraZeneca	64 (15)
Moderna	19 (5)
Vaccine type	
mRNA	349 (85)
Vector	64 (15)
Dose	
1st	156 (38)
2nd	257 (62)
Axillary symptom	
Yes	8 (2)
No	405 (98)
Axillary lymphadenopathy	
Yes	202 (49)
No	211 (51)

Note: unless otherwise specified, data represent the number of patients and data in parentheses are percentages. * Data are presented as means ± standard deviation; data in parentheses represent the ranges.

**Table 2 jcm-11-00238-t002:** Comparison of characteristics between the lymphadenopathy and non-lymphadenopathy.

Characteristics	Lymphadenopathy (*n* = 202)	Non-lymphadenopathy (*n* = 211)	*p* Value
Mean age (years) *	44 ± 11 (17–79)	52 ± 10 (24–80)	<0.001
Age group			<0.001
≤40 years	67 (33)	28 (13)	
>40 years	135 (67)	183 (87)	
Sex			0.106
Women	200 (99)	203 (96)	
Men	2 (1)	8 (4)	
Mean interval (days) *	20 ± 14 (2–80)	32 ± 19 (1–82)	<0.001
Interval group			<0.001
1–14 days	77 (38)	44 (21)	
15–28 days	82 (41)	49 (23)	
29–42 days	29 (14)	63 (30)	
≥43 days	14 (7)	55 (26)	
Site			0.341
Left arm	188 (93)	201 (95)	
Right arm	14 (7)	10 (5)	
Vaccine brand			<0.001
Pfizer	182 (90)	148 (70)	
AstraZeneca	5 (3)	59 (28)	
Moderna	15 (7)	4 (2)	
Vaccine type			<0.001
mRNA	197 (98)	152 (72)	
Vector	5 (2)	59 (28)	
Dose			<0.001
1st	98 (48)	58 (28)	
2nd	104 (52)	153 (73)	
Axillary symptom			0.003
Yes	8 (4)	0 (0)	
No	194 (96)	211 (100)	

Note: unless otherwise specified, data represent the number of patients and data in parentheses are percentages. * Data are presented as means ± standard deviation; data in parentheses represent the ranges.

**Table 3 jcm-11-00238-t003:** Multiple linear regression analysis used to determine the risk factors associated with axillary lymphadenopathy.

Variables	Odds Ratio (95% CI)	*p* Value
Age group		
≤40 years	2.2 (1.3–3.8)	0.006
Interval group		
1–14 days	4.3 (2.0–9.4)	<0.001
15–28 days	4.3 (2.0–9.2)	<0.001
29–42 days	1.5 (0.7–3.4)	0.327
Vaccine type		
mRNA	6.7 (2.5–17.9)	<0.001
Dose		
1st	1.6 (1.0–2.5)	0.045
Axillary symptom		
No	0	>0.99

## Data Availability

Data are available on request from the corresponding author.

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
