# Peer review of "Axillary Lymphadenopathy on Ultrasound after COVID-19 Vaccination and Its Influencing Factors: A Single-Center Study"

_jcm, 2022, doi:10.3390/jcm11010238_

Round 1

Reviewer 1 Report

I believe that a good job has been done, in terms of actuality, interest and scientific value; the iconographic images are of good quality and the bibliography up to date.

Author Response

Reviewer 1

English language and style

(x) Extensive editing of English language and style required

Comments and Suggestions for Authors

I believe that a good job has been done, in terms of actuality, interest and scientific value; the iconographic images are of good quality and the bibliography up to date.

Response: We would like to thank for evaluating our manuscript and for your comment. Please note that we have sent our manuscript to an English editing company (Editage) for English proofreading. We hope that the language quality level has been significantly improved in the revised manuscript.

Reviewer 2 Report

This work adresses a very current topic. It's well written and well presented. It is of certain interest for the readers, except for missing infromation on follow up ultrasound, as the Athors themself point out. This would have certainly added significant value to the clinical relevance of this paper. Nevetheless only minor suggestion can be given:

- On page 3, lines 82-83, the Authors describe the method of image interpretation by the two radiologists involved in the study: the statement is not clear, please explain further. was is a double blind evaluation? what does it mean they interpreted images "in consensus"?

On page 3, lines 102-103 and 106 -107: is there a reason for not keeping consistent in indicating absolute numbers  followed by percentages between brackets? why did the Authors switch absolute numbers with percentages in and out of the brackets?

Author Response

Reviewer 2

English language and style

(x) English language and style are fine/minor spell check required

Comments and Suggestions for Authors

This work addresses a very current topic. It's well written and well presented. It is of certain interest for the readers, except for missing infromation on follow up ultrasound, as the Athors themself point out. This would have certainly added significant value to the clinical relevance of this paper. Nevetheless only minor suggestion can be given:

Response: The authors would like to thank the reviewer for your constructive critique to improve the manuscript. We have made every effort to address the issues raised and to respond to all comments. Please, find next a detailed, point-by-point response to the reviewer's comments. We hope that our revisions will meet the reviewer’s expectations.

- On page 3, lines 82-83, the Authors describe the method of image interpretation by the two radiologists involved in the study: the statement is not clear, please explain further. was is a double blind evaluation? what does it mean they interpreted images "in consensus"?

Response: We would like to thank the reviewer for the questions. Radiologists were blinded to the clinical information. They made a decision concerning the presence or absence of abnormal lymph nodes, after reaching to consensus when there were different opinions. We have revised the corresponding part in the manuscript as follows:

“The presence of abnormal axillar lymphadenopathy was retrospectively reviewed by two radiologists who were blinded to the patients’ clinical information. Abnormal lymphadenopathy was defined as enlarged lymph nodes with cortical thickening >3 mm with or without a preserved fatty hilum on ultrasound. Any discrepancies were resolved by consensus.” (Lines 82–86)

- On page 3, lines 102-103 and 106 -107: is there a reason for not keeping consistent in indicating absolute numbers followed by percentages between brackets? why did the Authors switch absolute numbers with percentages in and out of the brackets?

Response: We would like to thank the reviewer for the question. Please note that we have corrected the order of the numbers and percentages to ensure consistency, as per the reviewer’s suggestion.
